# Sex-Specific Asymmetrical Attack Rates in Combined Sexual-Vectorial Transmission Epidemics

**DOI:** 10.3390/microorganisms7040112

**Published:** 2019-04-25

**Authors:** Ana Carolina W. G. de Barros, Kaline G. Santos, Eduardo Massad, Flávio Codeço Coelho

**Affiliations:** Fundação Getulio Vargas, Escola de Matemática Aplicada, Rio de Janeiro 22250-9000, Brazil; acwgdb@gmail.com (A.C.W.G.d.B.); kaline96.st@gmail.com (K.G.S.); eduardo.massad@fgv.br (E.M.)

**Keywords:** epidemiology, ℛ_0_, vector-borne disease, sexually-transmitted disease

## Abstract

In 2015–2016, South America went through the largest Zika epidemic in recorded history. One important aspect of this epidemic was the importance of sexual transmission in combination with the usual vectorial transmission, with asymmetrical transmissibilities between sexual partners depending on the type of sexual contact; this asymmetry manifested itself in data as an increased risk to women. We propose a mathematical model for the transmission of the Zika virus including sexual transmission via all forms of sexual contact, as well as vector transmission, assuming a constant availability of mosquitoes. From this model, we derive an expression for $R_{0}$, which is used to study and analyze the relative contributions of the male to female sexual transmission route vis-à-vis vectorial transmission. We also perform Bayesian inference of the model’s parameters using data from the 2016 Zika epidemic in Rio de Janeiro.

## 1. Introduction

Vector-borne diseases and sexually-transmitted diseases have had their transmission dynamics extensively analyzed through mathematical models [1,2]. Models and therefore diseases combining both modes of transmission are less common, but it is particularly rare to have detailed data available to validate theoretical considerations raised by the modeling [3,4].

Zika virus (ZIKV) was originally isolated from samples from the Zika forest in Uganda obtained from a rhesus monkey in 1947 [5]. The first cases of human infection were recorded in Nigeria and Tanzania from 1952–1954 [6], spreading slowly across the Asian continent. Before 2007, ZIKV was not considered a disease of substantial concern to human beings because only isolated cases involving small populations had been reported worldwide [7]. ZIKV transmission had previously only been documented in regions of Africa and Asia, but in 2007, it was detected in Yap, Micronesia, causing a small outbreak. Forty-nine ZIKV-infected cases were confirmed. In 2012/2013, it caused a new outbreak in French Polynesia, spreading across the other Pacific islands, resulting in an epidemic with more than 400 confirmed cases [8].

In March of 2015, the virus was detected for the first time in Brazil [9]. In October 2015, a growing number of cases of newborns with microcephaly were reported in Pernambuco. In November, after confirmation of ZIKV presence in the amniotic fluid of pregnant women in the state of Paraíba, the association between the virus infection and microcephaly was confirmed [6].

ZIKV spread rapidly throughout Brazil, and in less than a year, it had reached the entire country, spreading to neighboring countries. It has been estimated that in 2015 alone, there were between 440,000 and 1,300,000 Zika cases in Brazil, resulting in the largest ZIKV epidemic reported so far [10].

Although ZIKV is a flavivirus transmitted to humans primarily through the bite of infected *Aedes* mosquitoes [11], it can also spread through sexual contact. Studies have shown the presence of a infectious viral load in the semen after Zika infection [12], indicating that we are dealing with a pathogen that is also sexually transmissible [13]. Although not very common, other forms of transmission between humans have been confirmed recently from the identification of virus in the urine and saliva [14]. The most reported and well-characterized form of sexual transmission among humans is from male to female. However, transmission through homosexual relations and from women to men have also been reported [15,16].

Zika transmission has been modeled mathematically before [17,18,19]. In this paper, we derive a closed-form expression for $R_{0}$ resulting from a model combining both sexual and vectorial transmissions.

Given the current understanding of the mechanisms behind the transmission of ZIKV and its consequences to human health, we expect our model to help in the investigation of the relative importance of the vectorial and sexual transmission to the dynamics of Zika.

Finally, we fit the model to data from the 2016 Zika epidemic in Rio de Janeiro to estimate some of its parameters and to demonstrate that this model is able to explain a real Zika epidemic with different burdens for women and men.

## 2. Methods

The proposed epidemic model used is an adaptation of an SEIRmodel proposed earlier [20]. We assume a constant population size, ignoring demographic processes such as birth, death, and immigration, as well as seasonal fluctuations. This is justified as we are studying the dynamics of the system during a single epidemic where demographic variations can be considered negligible.

At any given instant *t*, the state of the system is represented by the fractions of the total population N(t) in each of the immunological states described in Equation (1), further split by sex. The dynamics of the mosquito population is not explicitly included in the model being represented by a single parameter of vectorial transmission. This implies that during the epidemic, the mosquito populations is sufficiently large to provide a constant vector transmission rate.
(1a)$$\begin{array}{c} N(t)=W(t)+M(t) \end{array}$$
(1b)$$\begin{array}{c} W\left(t\right)=W_{S}\left(t\right)+W_{E}\left(t\right)+W_{I}\left(t\right)+W_{R}\left(t\right) \end{array}$$
(1c)$$\begin{array}{c} M\left(t\right)=M_{S}\left(t\right)+M_{E}\left(t\right)+M_{I}\left(t\right)+M_{L}+M_{R}\left(t\right) \end{array}$$

The susceptible female and male populations, $W_{S}\left(t\right)$ and $M_{S}\left(t\right)$, respectively, are exposed to the Zika virus with a $\beta_{S}$ sexual transmission rate and a $\beta_{V}$ vector transmission rate. Since we are not modeling the mosquito population dynamics, we simplify the vectorial transmission to a mass-action term, e.g., $\beta_{V}\left(M_{I}\left(t\right)+W_{I}\left(t\right)\right)W_{S}\left(t\right)$.

Due to the fact that the virus remains viable in the semen, men may go into a latent state, $M_{L}\left(t\right)$, indicating a longer sexual infectious period as a whole. However, since the latent period is not the same for all men, we add the parameter ρ, which takes into consideration the fact that not all men go into latency.

To account for different rates of sexual transmission between sexes, we take the male-to-female transmission rate to be the baseline, the most effective, due to the larger inoculum. We denote it by $\beta_{S}$ and define $k_{WW}$, $k_{WM}$, $k_{MM}$, and $k_{L}$ to be a ratio of this baseline transmission contributed by other modes of sexual transmission, respectively woman-to-woman, woman-to-man, man-to-man, and latent-man-to-woman. We further assume these ratios to range from 0–1, meaning the other modes can at most be as effective as the male-to-female transmission.

We assume permanent and perfect immunity against new ZIKV infections after the first infection; therefore, recovered individual are not included explicitly in the model.

The proposed model consists of a system of seven ordinary differential equations:
(2a)$$\begin{array}{c} \frac{dW_{S}}{dt}=-\beta_{S}\left(k_{WW}W_{I}\left(t\right)+M_{I}\left(t\right)+k_{L}M_{L}\left(t\right)\right)W_{S}\left(t\right)-\beta_{V}W_{S}\left(t\right)\left(M_{I}\left(t\right)+W_{I}\left(t\right)\right) \end{array}$$
(2b)$$\begin{array}{c} \frac{dW_{E}}{dt}=\beta_{S}\left(k_{WW}W_{I}\left(t\right)+M_{I}\left(t\right)+k_{L}M_{L}\left(t\right)\right)W_{S}\left(t\right)+\beta_{V}W_{S}\left(t\right)\left(M_{I}\left(t\right)+W_{I}\left(t\right)\right)-eW_{E}\left(t\right) \end{array}$$
(2c)$$\begin{array}{c} \frac{dW_{I}}{dt}=eW_{E}\left(t\right)-\mu W_{I}\left(t\right) \end{array}$$
(2d)$$\begin{array}{c} \frac{dM_{S}}{dt}=-\beta_{S}\left(k_{WM}W_{I}\left(t\right)+k_{MM}\left(M_{I}\left(t\right)+M_{L}\left(t\right)\right)\right)M_{S}\left(t\right)-\beta_{V}M_{S}\left(t\right)\left(M_{I}\left(t\right)+W_{I}\left(t\right)\right) \end{array}$$
(2e)$$\begin{array}{c} \frac{dM_{E}}{dt}=\beta_{S}\left(k_{WM}W_{I}+k_{MM}\left(M_{I}\left(t\right)+M_{L}\left(t\right)\right)\right)M_{S}\left(t\right)+\beta_{V}M_{S}\left(t\right)\left(M_{I}\left(t\right)+W_{I}\left(t\right)\right)-eM_{E}\left(t\right) \end{array}$$
(2f)$$\begin{array}{c} \frac{dM_{I}}{dt}=eM_{E}\left(t\right)-\mu M_{I}\left(t\right) \end{array}$$
(2g)$$\begin{array}{c} \frac{dM_{L}}{dt}=\rho \mu M_{I}\left(t\right)-\tau_{l}M_{L}\left(t\right) \end{array}$$

The model’s variables and parameters are described in Table 1.

### 2.1. Attack Rates by Sex

From the model defined by Equation (2), we can define another two differential equations to track the attack rates over time. In epidemiology, the attack rate is the ratio between the number of cases and the population at risk. It is usually calculated for the entire epidemic, but we can also calculate it up to each point in time since the beginning of the epidemic, from the solutions of the following equations for the accumulated number of cases.
(3a)$$\begin{array}{c} \frac{dC_{W}\left(t\right)}{dt}=eW_{E}\left(t\right) \end{array}$$
(3b)$$\begin{array}{c} \frac{dC_{M}\left(t\right)}{dt}=eM_{E}\left(t\right) \end{array}$$
Since we are working with normalized populations where N(t)=1,∀t with an equal number of men and women, the attack ratios for women and men are given by $AR_{W}\left(t\right)=2C_{W}\left(t\right)$ and $AR_{M}\left(t\right)=2C_{M}\left(t\right)$, respectively. The ratio $AR_{W}\left(t\right)/AR_{M}\left(t\right)$ will represent the female burden relative to men. To calculate this ratio, which is shown in Figure 5, we have fixed an epidemic duration of 120 days to accumulate cases. We chose 120 days because arbovirus epidemics are usually restricted by weather conditions affecting mosquito activity and longevity and rarely last longer than 120 days.

### 2.2. Sexual Force of Infection

As we are studying the impact of sexual transmission of Zika, it is worth taking a closer look at the sexual force of infection. The force of infection in epidemic models is typically defined as the product of the transmissibility parameter and the fraction of infectious individuals in the populations. Here, since infection can happen both through direct sexual contact and vector transmission, it makes sense to distinguish between a sexual force of infection and a vectorial one. Let us first define the sexual force of infection for each sex, since their exposure to sexual infection is different. Let $\lambda_{SW}$ denote the force of infection of women and $\lambda_{SM}$ the force of infection of men.
(4)$$\begin{array}{cc} \lambda_{SW}\left(t\right) & =\beta_{S}\left(k_{WW}W_{I}+M_{I}+k_{L}M_{L}\right) \end{array}$$
(5)$$\begin{array}{cc} \lambda_{SM}\left(t\right) & =\beta_{S}\left(k_{WM}W_{I}+k_{MM}\left(M_{I}+M_{L}\right)\right) \end{array}$$

If we assume that transmission from men to women is the only relevant form of sexual transmission ($k_{WW}=k_{WM}=k_{MM}=0$), we can simplify the sexual forces of infections above to $\lambda_{SW}=\beta_{S}\left(M_{I}+K_{L}M_{L}\right)$ and $\lambda_{SM}=0$.

We can also define a vectorial force of infection in a similar fashion: $\lambda_{V}=\beta_{V}\left(M_{I}+W_{I}\right)$. Then, we can rewrite our model as:
(6a)$$\begin{array}{c} \frac{dW_{S}}{dt}=-\left(\lambda_{SW}+\lambda_{V}\right)W_{S} \end{array}$$
(6b)$$\begin{array}{c} \frac{dW_{E}}{dt}=\left(\lambda_{SW}+\lambda_{V}\right)W_{S}-eW_{E} \end{array}$$
(6c)$$\begin{array}{c} \frac{dW_{I}}{dt}=eW_{E}-\mu W_{I} \end{array}$$
(6d)$$\begin{array}{c} \frac{dM_{S}}{dt}=-\left(\lambda_{SM}+\lambda_{V}\right)M_{S} \end{array}$$
(6e)$$\begin{array}{c} \frac{dM_{E}}{dt}=\left(\lambda_{SM}+\lambda_{V}\right)M_{S}-eM_{E} \end{array}$$
(6f)$$\begin{array}{c} \frac{dM_{I}}{dt}=eM_{E}-\mu M_{I} \end{array}$$
(6g)$$\begin{array}{c} \frac{dM_{L}}{dt}=\rho \mu M_{I}-\tau_{l}M_{L} \end{array}$$

### 2.3. Epidemic Threshold and $R_{0}$

One of the central questions in mathematical epidemiology is about the ability of a disease to invade a population, i.e., generate an epidemic. The answer to this question is related to the stability of the state we call disease-free-equilibrium (DFE), which is basically the state in which the number of infectious individuals is zero ($M_{I}+W_{I}=0$ and $\frac{dW_{I}}{dt}=\frac{dM_{I}}{dt}=0$). Here, we will calculate the $R_{0}$ of the model, which is the expected number of secondary cases generated by the introduction of a single infectious individual when the system is at the DFE. The spectral radius (the largest eigenvalue, in absolute value) of the next-generation matrix defined at the DFE is $R_{0}$. We define the epidemic threshold as $R_{0}=1$, because the disease will be able to invade the population only when $R_{0}>1$. We show how to construct the next-generation matrix and calculate $R_{0}$ in the Results Section.

### 2.4. Fitting the Model to Data

In order to approximate the relative relative contributions of sexual and vectorial transmission in the context of the proposed model, we fitted the model to the observed incidence of Zika in men and women in the 2016 epidemic in Rio de Janeiro [14].

We calculated the prevalence of Zika for both sexes by dividing the weekly-reported number of cases by the population size for each sex multiplied by the underreporting rate estimated for that epidemic [21].

To fit the model to data, we applied the Bayesian inference methodology proposed by Coelho et al. [22]. We chose to estimate the parameters listed in Table 2, keeping the remaining parameters fixed. The prior distributions chosen for these parameters are listed in Table 2.

## 3. Results

We performed numerical simulations of the dynamics based on parameter values obtained from the literature. For the parameters for which no experimental measurements were available, we explored the ranges of values described in Table 1.

Figure 1 shows one of the simulations where it is interesting to note the higher prevalence of both exposed and infectious women in the first 2–3 months of the epidemic.

### 3.1. Sexual Force of Infection

Due to its dependence on the prevalence of infectious men in the population, the sexual force of infection in women displayed a quite different profile, as shown in Figure 2, in comparison to the vectorial force of infection.

The effects of the sexual force of infection in terms of how effective the sexual transmission is in the post-viremic stage ($K_{L}$) can be seen in Figure 3. The extra source of risk to women, associated with transmission from infectious men (both acute and latent), is evident as well in Figure 5, which makes it clear that in the presence of sexual transmission, $\beta_{s}>0$, attack ratios for women will always be greater than for men.

Figure 4 depicts the qualitative difference in prevalence dynamics between the effects of underreporting and sexual transmission; the underreported curve for men is obtained by applying a constant 50% reporting rate to the men’s prevalence curve, $M_{I}$.

### 3.2. The Basic Reproduction Number: $R_{0}$

A very important parameter in epidemiology is the basic reproduction number or $R_{0}$ of the disease, which can be derived from the transmission model. It determines the epidemic potential of a transmissible disease. It is the average number of infections an infected individual is capable of producing when introduced to a completely susceptible population.

We derive the basic reproduction number for our model by means of the next generation matrix method [24]. According to this method, first we need to distinguish new infections from all the other changes in the population. Then, we let *m* denote the number of compartments containing infected individuals. They are $W_{E}$, $W_{I}$, $M_{E}$, $M_{I}$, and $M_{L}$, so m=5. For clarity, we will order the n=7 compartments as: $\left[W_{E},W_{I},M_{E},M_{I},M_{L},W_{S},M_{S}\right]$, separating the m=5 first compartments from the rest. Then, we define the vector F[i] as the rates of appearance of new infections of new infections at each compartment *i*, i=1,…,m, when the system is in a disease-free state. It is worth pointing out that the transfers between exposed to infected and latent (in male cases) are not considered new infections, but the progression of an infected individual through many compartments. Likewise, we define the vector V[i] as the net flow of individuals in and out of the *m* compartments by other means. Therefore,
$$F=\left[\begin{array}{c} \frac{\left(M_{I}+M_{L}k_{L}+W_{I}k_{WW}\right)M_{S}\beta_{s}}{\mu }+\frac{\left(M_{I}+W_{I}\right)W_{S}\beta_{v}}{\mu }\\ 0\\ \frac{\left(\left(M_{I}+M_{L}\right)k_{MM}+W_{I}k_{WM}\right)M_{S}\beta_{s}}{\mu }+\frac{\left(M_{I}+W_{I}\right)M_{S}\beta_{v}}{\mu }\\ 0\\ 0 \end{array}\right]\;V=\left[\begin{array}{c} eW_{E}\\ -eW_{E}+\mu W_{I}\\ eM_{E}\\ -eM_{E}+\mu M_{I}\\ \tau_{l}M_{L} \end{array}\right]$$

The next step is to calculate the Jacobian of each matrix above, to obtain *F* and *V* for the disease-free equilibrium solution: $W_{I}=0$, $M_{I}=0$, $W_{S}=1/2$, $M_{S}=1/2$, $W_{E}=0$, $M_{E}=0$, and $M_{L}=0$.
$$F=\left[\begin{array}{ccccc} 0 & \frac{\beta_{s}k_{WW}}{2}+\frac{\beta_{v}}{2} & 0 & \frac{\beta_{s}}{2}+\frac{\beta_{v}}{2} & \frac{\beta_{s}k_{L}}{2}\\ 0 & 0 & 0 & 0 & 0\\ 0 & \frac{\beta_{s}k_{WM}}{2}+\frac{\beta_{v}}{2} & 0 & \frac{\beta_{s}k_{MM}}{2}+\frac{\beta_{v}}{2} & \frac{\beta_{s}k_{MM}}{2}\\ 0 & 0 & 0 & 0 & 0\\ 0 & 0 & 0 & 0 & 0 \end{array}\right]\;V=\left[\begin{array}{ccccc} e & 0 & 0 & 0 & 0\\ -e & \mu & 0 & 0 & 0\\ 0 & 0 & e & 0 & 0\\ 0 & 0 & -e & \mu & 0\\ 0 & 0 & 0 & -\mu \rho & \tau_{l} \end{array}\right]$$

The next generation matrix for the model is given by the product $FV^{-1}$, and $R_{0}$ is the spectral radius of the resulting matrix.
(7)$$\begin{array}{cc} R_{0} & =\frac{\beta_{s}k_{\mathrm{MM}}\mu \rho +\left(\beta_{s}k_{\mathrm{MM}}+\beta_{s}k_{\mathrm{WW}}+2\beta_{v}\right)\tau_{l}}{4\;\mu \tau_{l}}\\ & +\frac{\sqrt{\beta_{s}^{2}k_{\mathrm{MM}}^{2}\mu^{2}\rho^{2}+2\beta_{s}\left(\beta_{s}\left(k_{\mathrm{MM}}^{2}+2k_{L}k_{\mathrm{WM}}-k_{\mathrm{MM}}k_{\mathrm{WW}}\right)+2\beta_{v}k_{L}\right)\mu \rho \tau_{l}+\left(\beta_{s}^{2}\left(k_{\mathrm{MM}}^{2}-2k_{\mathrm{MM}}k_{\mathrm{WW}}+k_{\mathrm{WW}}^{2}\right)+4\left(\beta_{s}\beta_{v}+\beta_{v}^{2}+\beta_{s}\left(\beta_{s}+\beta_{v}\right)\right)k_{\mathrm{WM}}\right)\tau_{l}^{2}}}{4\;\mu \tau_{l}} \end{array}$$

If we disregard Zika sexual transmission between individuals of the same sex and from women to men, we get a simpler expression for $R_{0}$:(8)$$R_{0}^{*}=\frac{\beta_{v}+\sqrt{\frac{\beta_{s}\beta_{v}k_{L}\mu \rho }{\tau_{l}}+\beta_{s}\beta_{v}+\beta_{v}^{2}}}{2\;\mu }$$

We can also look to the basic reproduction number without sexual transmission, $\beta_{s}=0$:$$R_{0}^{v}=\frac{\beta_{v}}{\mu }$$

Due to the natural deviation of sexual transmission from a basic mass-action contact rate, as stated in the model, we should apply a correction to $R_{0}^{*}$ to accommodate the distribution of the number of sexual partners men have (since we are only considering sexually-transmitted infections). According to Anderson and May [25] (Chapter 11, page 233), this correction factor *c* is given by the expression $c=m+s^{2}/m$, where *m* is the mean number of sexual partners in the population and $s^{2}$ is its variance. We found the value of c=2.154 using male partner distribution data (m=1.339, $s^{2}=1.0917$) from a sexual behavior survey. All simulations were done with this correction.

Based on the reduced $R_{0}^{*}$ derived from the model, we investigated numerically the importance of the sexual transmission depending on attack ratio for men, $AR_{M}$, and women, $AR_{W}$, given different magnitudes of $\beta_{s}$ and $\beta_{v}$. For these simulations, we disregarded homosexual transmission and from women to men, assuming their contributions are minimal. In Figure 5, we can see the ratio $\frac{AR_{W}}{AR_{M}}$ as a function of the two modes of transmission. The green line represents $R_{0}^{*}=1$ calculated from Equation (7). We can see that for more intense values of sexual transmission and moderate vector transmission levels, the total number of female cases is much larger than male cases.

### 3.3. Fitting the Model to Data

We fitted the model from Equation (6). Table 2 contains the details about the parameters estimated. The remaining parameters were kept constant at these values: $k_{L}=1$, e=3.18, ρ=1, and $k_{MM}=k_{MH}=k_{HH}=0$. In Figure 6, we can see the joint posterior distribution of $\beta_{s}$ and $\beta_{v}$, and note that it roughly follows the relationship between these parameters depicted in Figure 7. In Figure 8, we can find the posterior boxplot for all parameters estimated. For the inference, a new parameter, ur, was added, corresponding to the underreporting of Zika in men relative to women: $M_{I}^{o}\left(t\right)=ur\times M_{I}\left(t\right)$ with $M_{I}^{o}\left(t\right)$ denoting the observed number of male cases. The posterior distributions of the prevalence series are shown in Figure 9.

The results point to a roughly a three-times higher transmissibility through sexual contact than via mosquito bites. We also see that about 20% of male cases go unreported.

## 4. Discussion

We propose here a mathematical model for studying the combined vectorial and asymmetrical sexual transmission of Zika. The model accommodates all known forms of Zika sexual transmission, as well as vectorial transmission. Vectorial transmission is simplified in the proposed model, because the model seeks to represent a single epidemic where mosquito density is high and is not likely to pose any limits to transmissibility. For modeling of multi-year dynamics, seasonal dynamics of the mosquito population would be required for an accurate description of the vectorial force of infection.

Due to the current lack of knowledge about the relative effectiveness of other forms of sexual transmission, such as from men to men and from women to women, we have limited our analysis to the better understood transmission from men to women. This lack of a more complete knowledge of the ZIKV transmission cycle is the main limitation of our results. Once the actual transmissibilities of all modes of transmission are known, as well as the relative prevalence in the population of each form of sexual contact, we will be able to exploit the full potential of this model. It is worth mentioning that the transmissibility-modifying parameters, as well as $\beta_{s}$ accommodate the fact that only young adults effectively contribute to sexual transmission [14]; thus, it is not necessary to represent sexually-active males and females as separate state variables in this model.

We have found that the asymmetrical nature of sexual transmission of ZIKV can lead to a larger burden on women when compared to men (Figure 1), simply due to the modified dynamics of transmission. Figure 5 shows the ratio $AR_{W}\left(120\right)/AR_{M}\left(120\right)$ in terms of the intensities of both sexual and vectorial transmission. This effect has been observed in Zika incidence data [14,17,26,27], but it was frequently confounded with gender-related underreporting; however, from our estimates of the male underreporting, we conclude that underreporting can account for only a 20% drop, with the rest of the difference being explained by the excess in female cases due to sexual transmission.

This is true even when $R_{0}<1$, as illustrated in Figure 5. Another interesting consequence of asymmetrical sexual transmission, i.e., it is easier for men to infect women than vice-versa, is that it allows us to differentiate its effects from those of a mere sexual bias in case reporting (it is known that women are more likely to report any illness than men), because the resulting dynamics are qualitatively different (Figure 4). It may be difficult to differentiate these effects from noisy incidence data, but they are nevertheless different things and should be treated accordingly.

Figure 7 shows that for the reduced model, without homosexual and women-to-men transmission, vectorial transmission is necessary to allow crossing the epidemic threshold of $R_{0}=1$. It follows that if some positive flow of virus is possible directly from men to men or from women to men, epidemics will be possible without vectorial transmission, but although there are reported cases of both of these kinds of transmission [15,16], no observational evidence is available so far of sustained transmission without vectors. The fact that the correction factor c=2.154 for male sexual contact heterogeneity is greater than one gives more importance to sexual transmission. Since men that have sex with men usually have substantially more sexual partners [28], the amplification of the sexual transmission of Zika in this community can be quite large if this mode of transmission is included. Moreover, with effective male-to-male transmission, sustained transmission of ZIKV becomes possible without the vector.

In Figure 6, the joint posterior distribution for the transmissibility parameters $\beta_{s}$ and $\beta_{v}$ shows a clear bimodality, meaning that if we could determine more precisely the range of values for $\beta_{v}$, which is easier to measure, this would lead to a more precise estimate of $\beta_{s}$.

Another important factor in the determination of the long-term dynamics of Zika in a population is the parameter $K_{L}$, which represents how capable men are at sexually transmitting ZIKV after the viremic period (Figure 3), when compared to during it. It has been confirmed that some men can test positive for ZIKV RNA in the semen months after the viremic period [29,30]. Our model shows that until we can properly measure or estimate the sexual transmissibilities between humans in the post-viremic stage, we must be prudent [31] and recommend protected sex to men returning from endemic areas with or without symptoms.

## Figures and Tables

**Figure 1 microorganisms-07-00112-f001:**
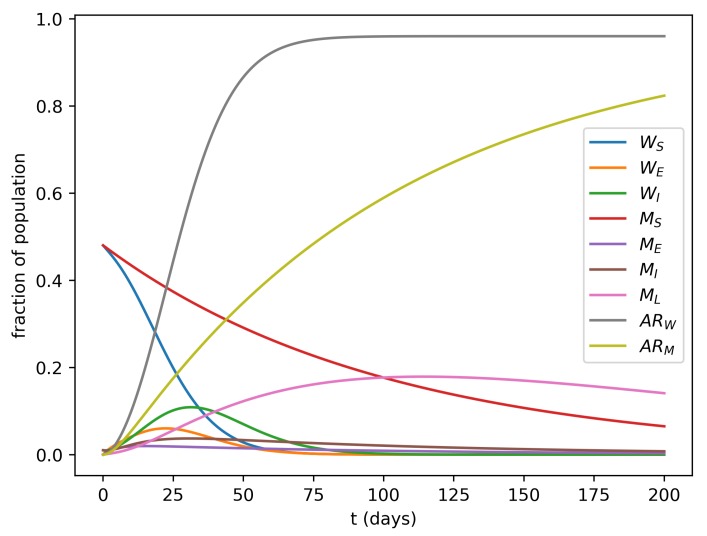
Simulation of the model’s dynamics with $\beta_{s}=0.25$, $\beta_{v}=0.01$, μ=0.1, e=0.2, $\tau_{l}=0.01$, ρ=1, and $K_{L}=1$. $R_{0}=3.04$, which is compatible with values reported by Villela et al. [23]. The $AR_{W}$ and $AR_{W}$ curves correspond to the attack rates over time for women and men, respectively.

**Figure 2 microorganisms-07-00112-f002:**
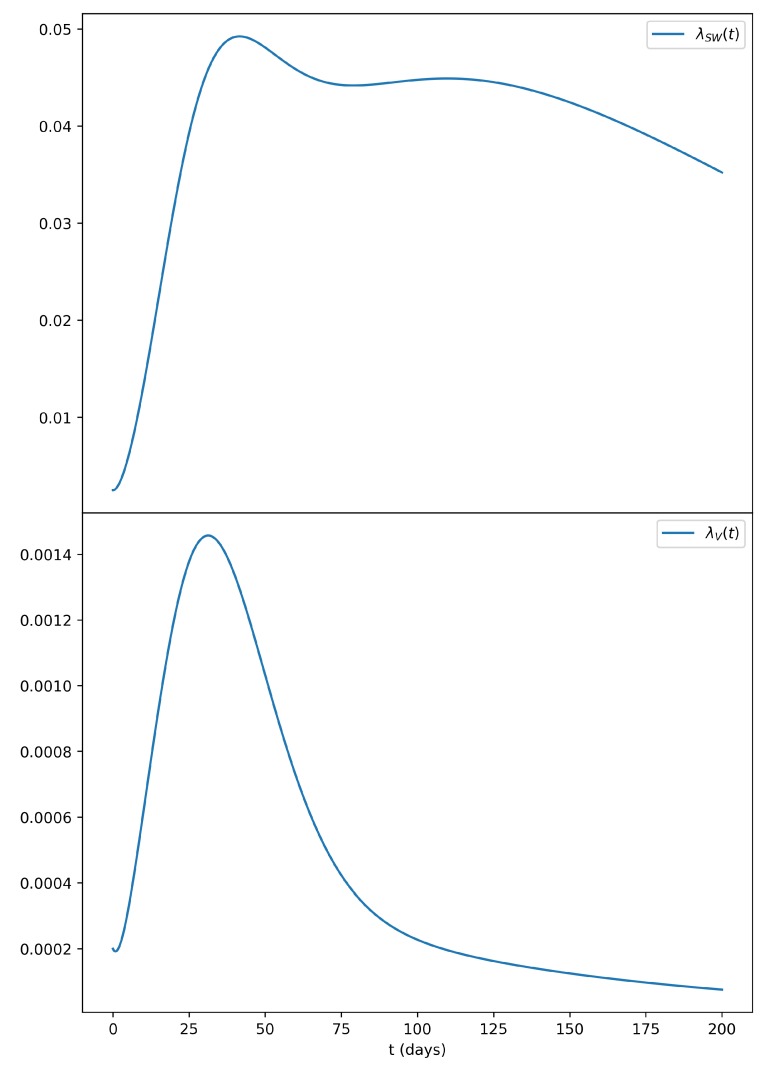
Sexual force of infection for women with the same parameters as those of Figure 1. In the top panel, we can observe that the sexual force of infection of women ($\lambda_{SW}\left(t\right)$) remains elevated for quite a longer period of time if compared to the vectorial force of infection ($\lambda_{V}\left(t\right)$) shown in the lower panel.

**Figure 3 microorganisms-07-00112-f003:**
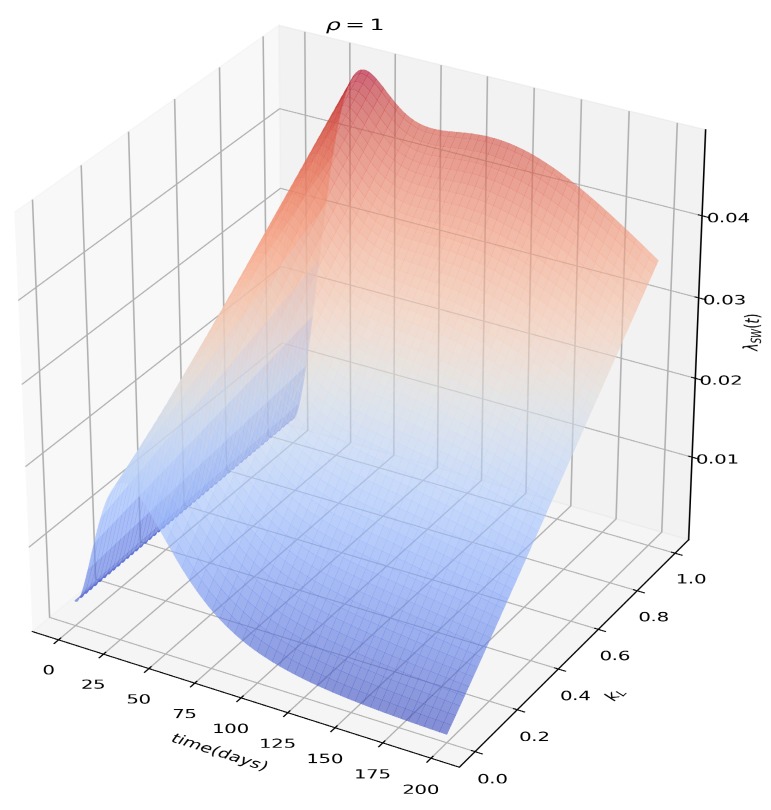
Sexual force of infection as a function of $k_{L}$, or how effective the sexual transmission from men to women is in the post-viremic phase. Notice that in the absence of effective longer term sexual transmission from men to women, the dynamics reverts to that of a standard vector-borne infection.

**Figure 4 microorganisms-07-00112-f004:**
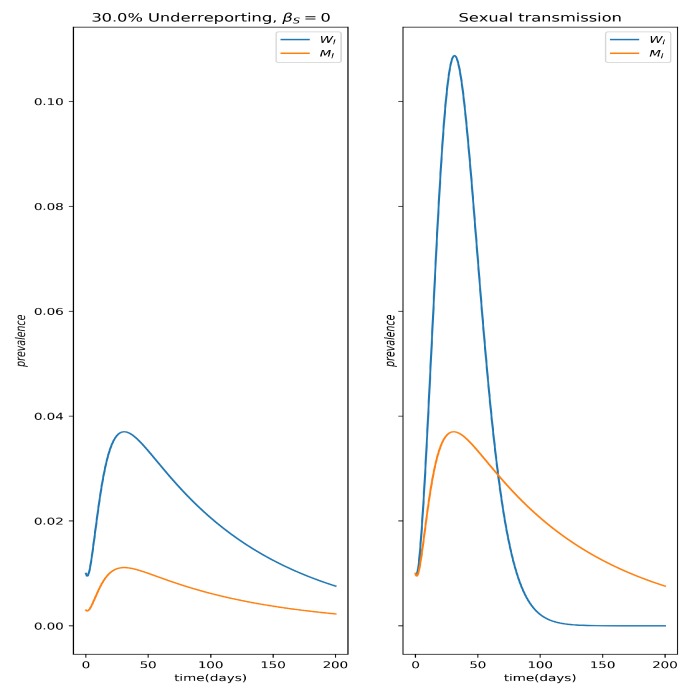
Qualitative differences between the impact of sexual bias in reporting, namely underreporting of male cases (**left** panel) and sexual transmission in the prevalence curves $W_{I}\left(t\right)$ and $M_{I}\left(t\right)$ (**right** panel). Notice that the crossing of the prevalence curves indicates the presence of sexual transmission as this can never happen from underreporting alone.

**Figure 5 microorganisms-07-00112-f005:**
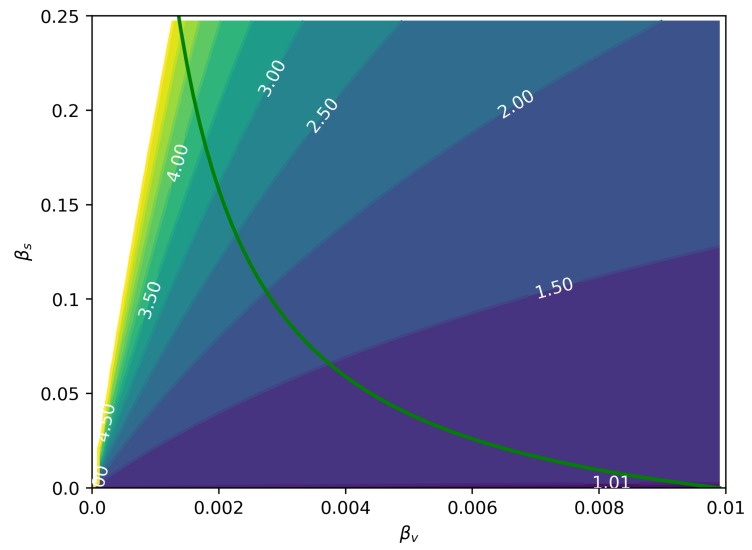
Ratio $\frac{AR_{W}\left(120\right)}{AR_{M}\left(120\right)}$ for a range of $\beta_{s}$ and $\beta_{v}$ values. The green line represents $R_{0}=1$, i.e., the epidemic threshold. Any point to the right of this curve has $R_{0}>1$. It is worth noticing that the reported excess cases reported for Zika in women are possible both during epidemics and off-season [26].

**Figure 6 microorganisms-07-00112-f006:**
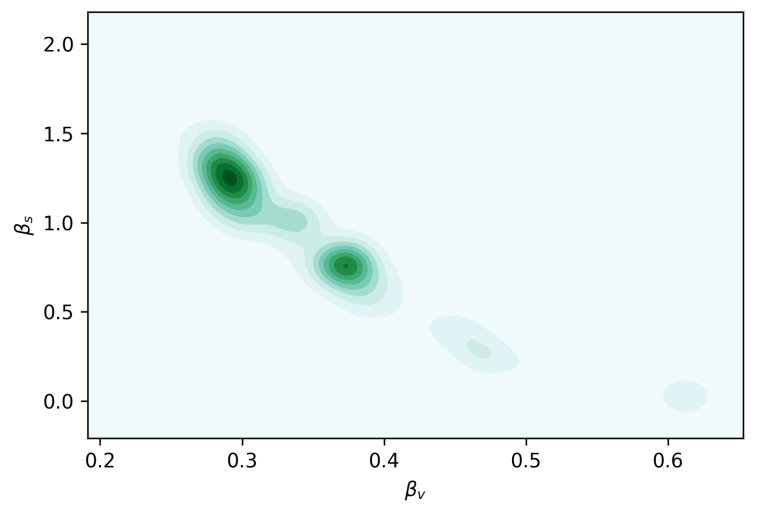
Joint posterior distribution of the transmission parameters $\beta_{s}$ and $\beta_{v}$.

**Figure 7 microorganisms-07-00112-f007:**
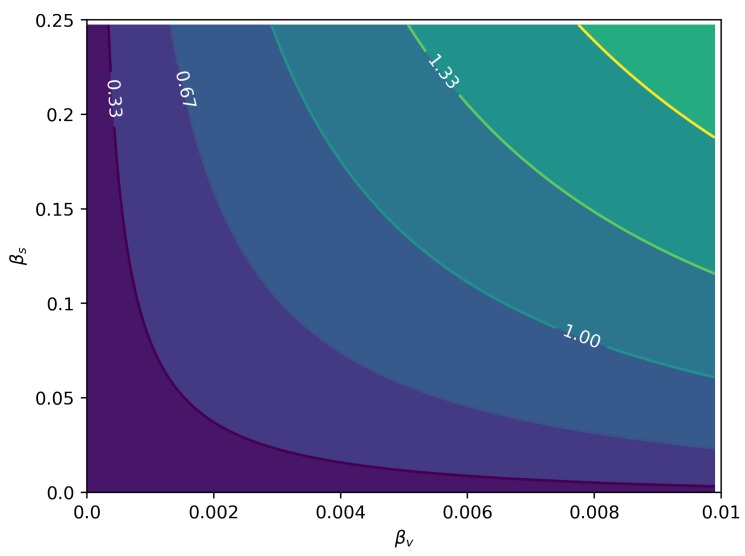
$R_{0}$ as a function of the relative intensities of sexual ($\beta_{s}$) and vectorial ($\beta_{v}$) transmissions. The $R_{0}$ values are already adjusted for the heterogeneity in sexual contact rates.

**Figure 8 microorganisms-07-00112-f008:**
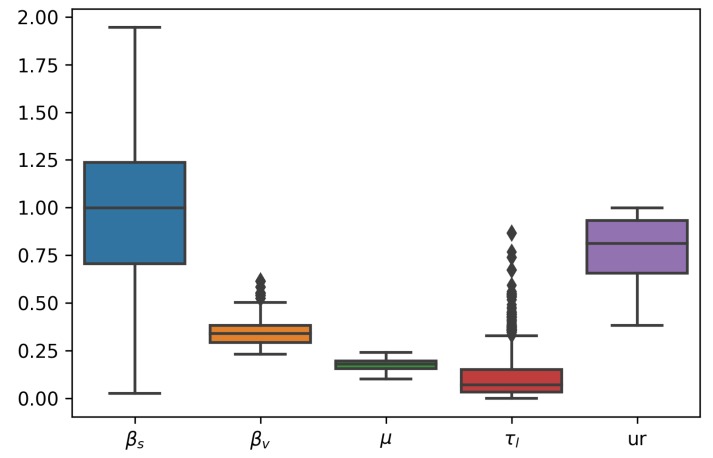
Boxplot of posterior distributions for all parameters estimated.

**Figure 9 microorganisms-07-00112-f009:**
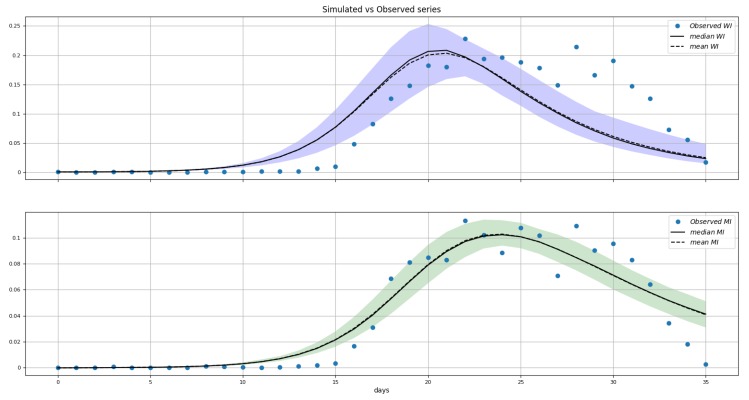
Posterior distributions of $W_{I}\left(t\right)$ (top panel) and $M_{I}\left(t\right)$ (bottom panel). Shaded areas represent 95% credibility intervals. Blue dots are the data. The Y-axis is the prevalence as a fraction of the population.

**Table 1 microorganisms-07-00112-t001:** Variables and parameters of the model. Values obtained from the literature are marked with references numbers. Ranges marked with a ${}^{†}$ correspond to values explored in simulations, but for which no experimental data could be found. * Fraction of the entire population, *N*.

Symbols	Description	Value Range
$W_{S}$	Susceptible women	[0,1] *
$W_{E}$	Exposed women	[0,1] *
$W_{I}$	Infectious women	[0,1] *
$M_{S}$	Susceptible men	[0,1] *
$M_{E}$	Exposed men	[0,1] *
$M_{I}$	Infectious men	[0,1] *
$M_{L}$	Latent men	[0,1] *
μ	Recovery rate (day${}^{-1}$)	[0.001,0.1] [18]
$\beta_{V}$	Vector transmission rate (day${}^{-1}$)	[0.1,0.75] [18]
$\beta_{S}$	Sexual transmission rate ((people × day)${}^{-1}$)	$\left[0,2\right]{\;}^{†}$
$k_{WW}$	Women-to-women transmissibility modifier	$\left[0,1\right]{\;}^{†}$
$k_{WM}$	Women-to-men transmissibility modifier	$\left[0,1\right]{\;}^{†}$
$k_{MM}$	Men-to-men transmissibility modifier	$\left[0,1\right]{\;}^{†}$
$k_{L}$	Latent period transmissibility modifier	$\left[0,1\right]{\;}^{†}$
*e*	Incubation rate (day${}^{-1}$)	[0.14,0.5] [18]
ρ	Fraction of men becoming latent	$\left[0,1\right]{\;}^{†}$
$\tau_{l}$	Latent recovery rate (day${}^{-1}$)	[0.025,0.1] [12]

**Table 2 microorganisms-07-00112-t002:** Parameters estimated from data, along with prior and posterior distributions. ${}^{†}$ The exponential distribution is parameterized in standardized form (loc, scale). Posteriors are given as medians and the 95% credible interval. All remaining parameters were kept constant.

Parameter	Prior	Posterior
$\beta_{s}$	N(0.98,0.3)	0.99[0.02,1.54]
$\beta_{v}$	N(0.25,0.1)	0.33[0.24,0.61]
μ	N(0.15,0.1)	0.17[0.10,0.23]
$\tau_{l}$	$Exp\left(0.0001,0.1\right){\;}^{†}$	0.7[0.005,0.37]
ur	U(0,1)	0.81[0.44,0.99]

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
