# Peer review of "Sex-Specific Asymmetrical Attack Rates in Combined Sexual-Vectorial Transmission Epidemics"

_microorganisms, 2019, doi:10.3390/microorganisms7040112_

Reviewer 1 Report

The paper is interesting, the argument is of interest of the journal. However the presentation is a little bit technical and without explanation of the calculation made. For example the technique of the next generation matrix needs some explanation. The spectral radius is connected to the eigenvalue. Explain how this is connected with a stable equilibria etc.Moreover is not clear what parameters have been computed with Bayesian techniques, and how the model has been fitted to experimental data. The main point is the asymmetry in sexual transmission, but is not clear how this come out.

I suggest to move the calculation in a appendix and to put in the appendix more deteils on the used approach.

Author Response

Regarding the improvements to the methods section particularly to the explanation of the R0 and the Next generation matrix, we have added a subsection to the methods to explain in more details the relationship between R0 and the stability of a disease-free-equilibrium, and the spectral radius of the Next generation matrix.

we have also extended the explanation of the Bayesian inference procedure in the Methods section.

We have also expanded the results subsection for "Sexual force of infection" clarifying the risk asymmetry causal relation with the directional differences in sexual transmission.

Reviewer 2 Report

The manuscript "Sex-specific asymmetrical attack rates in combined sexual-vectorial transmission epidemics" presents a ODE model that emphasizes sexual transmission over vector transmission in ZIKA.

Although the model and the associated estimations seem free of error, it is difficult to buy that a larger transmission rate was due to sexual encounters than to vectors, as expressed in figure 5. Of course, this is hard to tease apart given that mosquito dynamics are not explicitly presented, but authors should at least try to caution about the limitations of the model regarding that inferences are constrained mainly to the phenomenology of sexual transmission, since, at the end, vector transmission enters as a parameter for the transmission between infected human hosts via a vector, and not a vector and a human host, and, for example, it is not clear if this should have been done without delays when compared with sexual transmission. Similarly, although the model was fitted to the Rio de Janeiro data, it was done in a way where it was assumed all cases occurred in a sexually active population. Again, these are important modelling assumptions that bring strong limitations about model inference and abstractions that are not mentioned anywhere in the manuscript.

Other essential minor comments to be considered are presented in the commented pdf attached to this review.

Author Response

We have included citations to all the recommended articles about models of mosquito-borne transmission in the first paragraph of the introduction.

We have applied all the text corrections suggest on page 1

Regarding the remark about the simplification of the vectorial transmission present in the model, we have added an explanation in first paragraph of the discussion, stating that ignoring population dynamics is only acceptable because we are modeling a single epidemic, and that during epidemic season mosquito abundance does not vary much.

Regarding the comment that the model seems to imply that the entire population is sexually active, our response is that we acknowledge that only a fraction of the population is sexually active and we have described that in previous publications(refs 14 and 21 of the revised manuscript), However, this factor does not need to be included explicitly in the model as it can be expressed in the transmissibility modifier parameters and would not affect the qualitative dynamics of the model.  But we added this explanation to the second paragraph of the discussion.